# Quantifying protocols for safe school activities

**Juliano Genari** [1], **Guilherme Tegoni Goedert** [2,3,4], **Sérgio H. A. Lira** [5],
**Krerley Oliveira** [5], **Adriano Barbosa** [6], **Allysson Lima** [5], **José Augusto Silva** [5],
**Hugo Oliveira** [5], **Maurício Maciel** [5], **Ismael Ledoino** [7], **Lucas Resende** [8], **Edmilson Roque dos Santos** [1], **Dan Marchesin** [8], **Claudio J. Struchiner** [9,10], **Tiago Pereira** [1]*

**1** Instituto de Ciências Matemáticas e Computação, Universidade de São Paulo, São Paulo, São Paulo, Brazil, **2** Dipartimento di Fisica, Università degli Studi di Roma Tor Vergata and INFN, Rome, Lazio, Italy, **3** Aachen Institute for Advanced Study in Computational Engineering Science, RWTH AACHEN University, Aachen, North Rhine-Westphalia, Germany, **4** Computation-based Science and Technology Research Center, The Cyprus Institute, Nicosia, District of Nicosia, Cyprus, **5** Universidade Federal de Alagoas, Maceió, Alagoas, Brazil, **6** Universidade Federal de Grande Dourados, Dourados, Mato Grosso do Sul, Brazil, **7** Laboratório Nacional de Computação Científica, Petrópolis, Rio de Janeiro, Brazil, **8** Instituto de Matemática Pura e Aplicada, Rio de Janeiro, Rio de Janeiro, Brazil, **9** Escola de Matemática Aplicada, Fundação Getúlio Vargas, Rio de Janeiro, Rio de Janeiro, Brazil, **10** Instituto de Medicina Social, Universidade do Estado do Rio de Janeiro, Rio de Janeiro, Rio de Janeiro, Brazil

☯ These authors contributed equally to this work.
* tiago@icmc.usp.br

**Data Availability Statement:** The source code for COMORBUSS is available in the repository https://gitlab.com/ggoedert/comorbuss under the licende AGPLv3. The version of the code used together with all required data, input files and simulation

## Abstract

By the peak of COVID-19 restrictions on April 8, 2020, up to 1.5 billion students across 188 countries were affected by the suspension of physical attendance in schools. Schools were among the first services to reopen as vaccination campaigns advanced. With the emergence of new variants and infection waves, the question now is to find safe protocols for the continuation of school activities. We need to understand how reliable these protocols are under different levels of vaccination coverage, as many countries have a meager fraction of their population vaccinated, including Uganda where the coverage is about 8%. We investigate the impact of face-to-face classes under different protocols and quantify the surplus number of infected individuals in a city. Using the infection transmission when schools were closed as a baseline, we assess the impact of physical school attendance in classrooms with poor air circulation. We find that (i) resuming school activities with people only wearing low-quality masks leads to a near fivefold city-wide increase in the number of cases even if all staff is vaccinated, (ii) resuming activities with students wearing good-quality masks and staff wearing N95s leads to about a threefold increase, (iii) combining high-quality masks and active monitoring, activities may be carried out safely even with low vaccination coverage. These results highlight the effectiveness of good mask-wearing. Compared to ICU costs, high-quality masks are inexpensive and can help curb the spreading. Classes can be carried out safely, provided the correct set of measures are implemented.

## Author summary

The World Bank-UNESCO-UNICEF report [1] estimates that learning losses from the COVID-19 pandemic could cost this generation $17 trillion dollars in lifetime earnings. Despite the surging pressure to keep schools open, many countries lack guidelines for safe

scripts is available under the Tag Paper_Maragogi_Schools. The full documentation of the COMORBUSS library is available via https://docs.comorbuss.org/ under the license CC BY-SA 4.0.

**Funding:** This work was supported by the Center for Research in Mathematics Applied to Industry (FAPESP grants 2013/07375-0), by the Royal Society London, by the Brazilian National Council for Scientific and Technological Development (CNPq; grants 301778/2017-5, 302836/2018-7, 304301/2019-1, 306090/2019-0, 403679/2020-6) and by the Serrapilheira Institute (Grant No. Serra-1709-16124). JG was financed in part by the Coordenação de Aperfeiçoamento de Pessoal de Nível Superior – Brasil (CAPES) grant 88887.685473/2022-00. GTG has received funding from the European Union Horizon 2020 research and innovation programme under the Marie Skłodowska-Curie grant agreement No 765048. ERS was supported by FAPESP grant 2018/10349-4. KO and SHAL acknowledge the project promat-maragogi and CJS acknowledges the financial support of CNPq and FAPERJ, DM received funding from the grants CNPq-306566/2019-2 and FAPERJ - E-26/202.764/2017. Research carried out using the computational resources of the Center for Mathematical Sciences Applied to Industry (CeMEAI) funded by FAPESP (grant 2013/07375-0).

**Competing interests:** The authors have declared that no competing interests exist.

school activities. Using the empirical transmission level for closed schools as a baseline, we quantify the impact of distinct non-pharmaceutical interventions (NPIs) on infection rates and different values of vaccine coverage. Strikingly, we show that classes can be kept safe, provided the correct wearing of good quality masks together with to the proper combination of other NPIs. In such scenarios, the increase in infections can be kept below 20% compared to suspending classes.

## Introduction

The educational system plays a fundamental role in the socio-intellectual development and mental health of children and adolescents. During the COVID-19 pandemic, the impact of school closures on society has been massive. UNESCO reported that, as of April 8, 2020, up to 188 countries closed schools nationwide. In developing countries, such as Brazil, nutritional wellbeing of children was put in jeopardy as families rely on the provision of school meals. And yet, in Brazil alone, schools remained closed full-time for 191 days in 2020 affecting 44.3 million children. However, given the frequent contact during a school day, the prevalence of mild symptoms in children and the role of school as a source of contacts bridging family nuclei, there is understandable concern that face-to-face classes could be driving uncontrolled spreading of the virus.

In view of the negative physical and mental consequences for students, together with the educational deficit imposed by school closings, the ECDC agency points out that measures of transmission mitigation are necessary for students to have a safe socialization and learning environment [2]. Thus a major concern is the assessment of mitigation protocols [3] to understand the impact of each measure within the school community.

Living in a household with a child who goes to school physically increases the risk of becoming infected by up to 38%. Similarly, teachers working in school are 1.8 times more likely to be infected than those working from home [4] and resuming face-to-face classes has been directly related to outbreaks [2]. Mitigation measures such as separating student groups, quarantining exposed students and professionals, wearing masks, maintaining adequate air ventilation, vaccinating risk groups, and monitoring case emergence, can all decrease the number of new cases [4–6].

Often, mitigation measures are put in place simultaneously, which makes it difficult to disentangle their individual impact on transmission from temporal case report datasets. The lack of infrastructure, personnel and lab equipment may also limit the use of these measures in developing countries, especially when they are based on resource intensive practices such as testing and subsequent contact tracing of cases. Thus, it becomes crucial to identify effective mitigation practices *a priori*.

Our aim is to assess quantitatively the effects of vaccination [7] and NPIs protocols and find effective protocols for school activities. Our study shows that classes can be kept open safely, provided that the correct combination of measures be adopted. Relying on a single measure is mostly not effective nor stable, but simple measures can go a long way when properly combined and implemented.

## Materials and methods

### Data collection

The city of Maragogi in Northeastern Brazil has 33,000 inhabitants [8] and is a representative of at least 40% of Brazilian cities in terms of income and demographics. Moreover, its

demography is also typical worldwide, being located above the 50% quantile in a sample of 28,372 North American cities and 41,000 global cities, using the `simplemaps` database [9, 10], see section 1 in S1 File for further details.

Through a partnership with the city of Marogogi, established since March 2020, we developed a Clinical Monitoring System to track and trial all severe acute respiratory syndrome patients, see description in section 2 in S1 File. We also geo-localized patients and integrated this information with public data to obtain the household socio-economical data and family clusters section 3 in S1 File. The data integration is illustrated in the upper left panel of Fig 1. For our study, we used the data from May 9th, 2020 to July 25th, 2020, consisting of 18 confirmed deaths and 119 hospitalizations. In this period 1722 tests were performed, namely 52

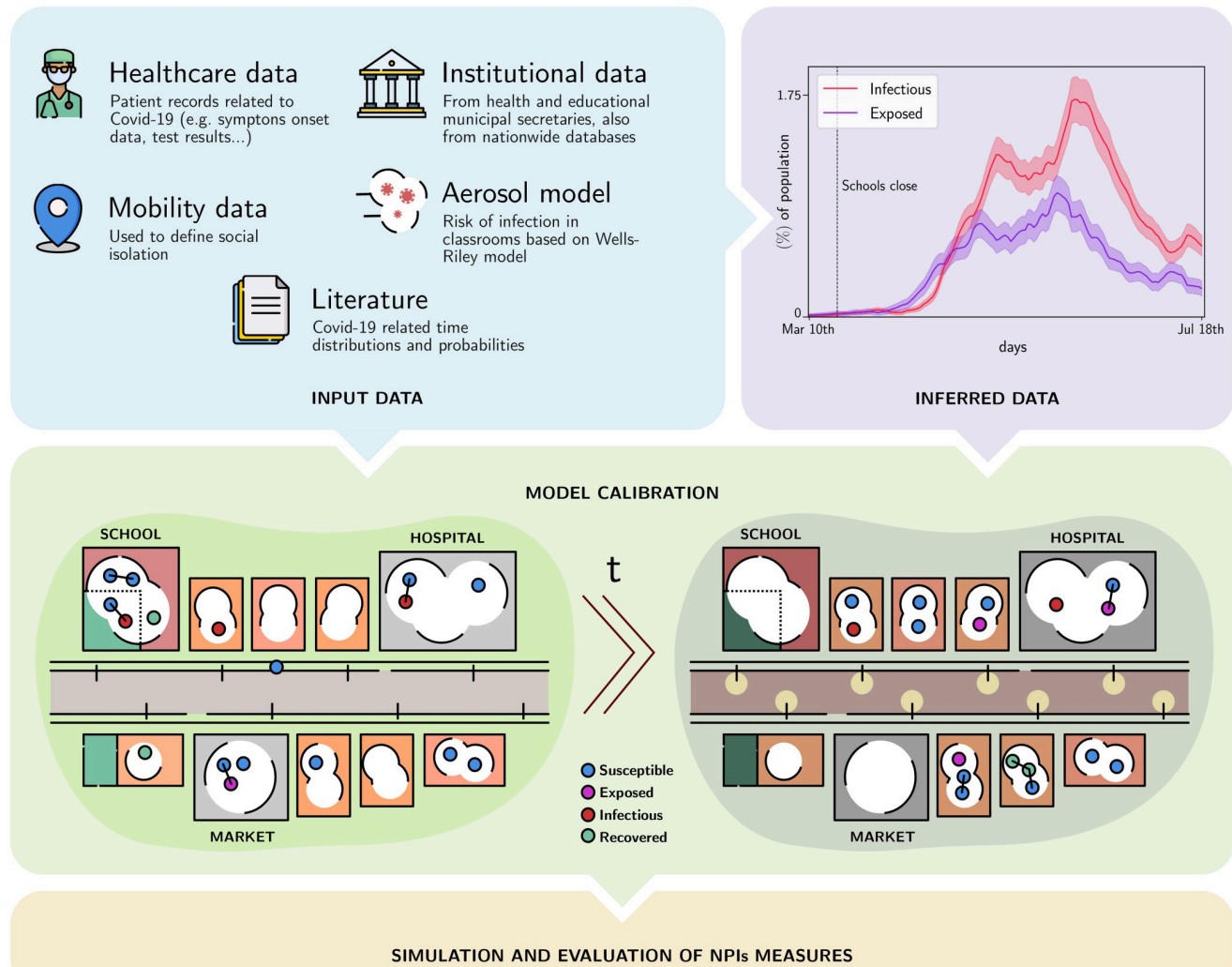

**Fig 1. Pipeline overview description.** Data is collected as patients attend health institutions. Health professionals register patients' personal, epidemiological and geo-location data to the Clinical Monitoring System (CMS) that is blended with socio-economical and household data. Using these data we estimate the number of Exposed (blue), Infectious (red) and Recovered (green) individuals. All the pre-processed data is used to calibrate our Stochastic Agent Based model, COMORBUSS. From bottom left to right: a schematic representation of the social dynamics of COMORBUSS, producing contacts between individuals in different social contexts. The colored circles represent the state of individuals, and lines represent relevant physical contacts capable of producing contagions. Once calibrated the model is used to estimate the effectiveness of NPIs.

RT-PRC tests and 1670 antibody tests (in majority COVID-19 IgG/IgM, see section 4 in S1 File for further details).

This study was approved by UFAL institutional Ethics Committee (CAAE: 43058821.9.0000.5013).

**Services.** We mapped the services that were allowed to be open during the period under government regulations and interviewed a sample of businesses for an estimate of daily occupation. The bulk of such services are food stores, building supply stores, restaurants and other minor retailer services as described in section 5 in S1 File.

**Street markets.** We estimated the usage of important open air services such as street markets by images collected via drones. We processed the images using the marking tool of the Drone Deploy mapping software [11] in order to evaluate the mean size and duration of cluster of people less than 2 meters apart during opening hours, as well as the average time spent by individuals in the street market. In Section 1 in S1 File, we also show that cities with demographics similar to Marogogi have analogous street market behavior.

**Health services.** During the period considered, the triage of all COVID-19 related cases was performed in a field hospital. We interviewed the staff of the health secretary to obtain data on the appointment mean time, and the mean number of contacts a patient has with doctors and other patients. This also provided data on the mean number of contacts among staff, see Section 5 in S1 File.

## Inference of states from data

We estimate the epidemiological SEIR curve from the attendance data of our Clinical Monitoring System. The SEIR curve corresponds to the trajectory of the population over the period of observation in the states: susceptible, exposed, infectious and recovered. The challenge is to transform the information of an individual reported in the attendance data into these states of the entire city population over time, correcting for sub-notification.

Under the hypothesis that all severe cases (hospitalization and death) are reported in our Clinical Monitoring System, for each reported individual we estimated the number of unreported infected individuals using a negative binomial (NB) distribution, and consequently, the total number of cases in the city over the period of observation. We modelled the total number of cases by $T = NB(p_h, 119) + 119$, where $p_h \approx 3.304\%$ is the estimated hospitalization probability for the city. We assume that these unreported individuals present their first symptoms at the same time as reported individuals.

Having all the individuals carrying the virus, we estimated how they progress across the SEIR compartments based on the severity of the case and the distribution of permanence of each state [12]. We rerun the statistical model 400 times to obtain SEIR curve samples for the city, see Section 4 in S1 File for further details. We denote by $\hat{v}$ the (empirical) distribution induced by these samples, e.g., the measure given by the uniform distribution over the 400 obtained samples.

## Agent based modeling

Agent based models are a class of computational models which track individual units (agents) of objects of interest. In the case of communal disease transmission, the natural choice for agents are the people which form that community and on whose contact the disease transmission is based. The two most important advantages of these models are: i) we can directly incorporate the biological and social heterogeneity of that community and investigate how it influences transmission patterns; ii) we are omniscient regarding the simulated histories of the

agents and can reliably evaluate the effects of specific public health protocols via conterfactual analysis of these histories.

Our agent based model, called COMORBUSS (COmmunitary Malady Observer of Reproduction and Behavior via Universal Stochastic Simulations), takes all these advantages a few steps further: we built a full model for the social dynamics of general communities in order to produce the contacts that drive disease propagation. We achieve this via a general modeling procedure of a city's infrastructure which can be systematically applied to any city via data integration. Moreover, our model is aware of the different roles the agents play in the various services that compose the infrastructure and produces contacts accordingly. This allows us to pinpoint the impact of a specific service and related mitigation protocols on disease spreading as well as track the resultant infection tree. To avoid overspecialized simulations of a single city, COMORBUSS stochastically produces for every simulation a realization of the transmission trajectory for the city in the class defined by the desired demographic and infrastructure data. For instance, each simulation has its own household network while satisfying the same distributions which describe household structure in that community. In the following, we describe the main pieces of the model and elaborate on its many details in Section 5 in S1 File. The most important parameters are classified and explained in Fig 2.

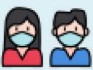

## Population Parameters

≫ `population_ages`: Number of citzens by age goup;
≫ `population_graph`: Samples of households structures (number of persons and respective ages);
≫ `persons_per_home`: Mean number of persons per household;

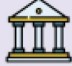

## Services Parameters

For each service modeled:

≫ `number`: Number of instances of this service in the city;
≫ `hours`: Openning and closing hours;
≫ `days`: Days of the week service opens;
≫ `visitation_period`: Mean time between visitations for each particle in the population;
≫ `isolation_visit_frac`: Factor to reduce visitations if an particle is in isolation;
≫ `net_par`: Configurations for the dynamic encounters network inside service;
≫ `workers`: Parameters to select workers and it's schedules, location inside service, etc;
≫ `inf_prob_weight`: Factor to apply to the probability of infections in this service;

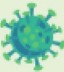

## Disease Parameters

≫ `inf_probability`: Probability of an infection given an encounter between an S and a I particle;
≫ `inf_prob_sympt` and `inf_severe_sympt_prob`: Probability by age goup of an infected particle to develop symptoms and severe symptoms;
≫ `inf_severe_death_prob`: Probability of an infection to end in death of the particle;
≫ `inf_duration` and `inf_severe_duration`: Mean time a particle stays infectious for normal and severe infections;
≫ `inf_incubation`: Mean time in the exposed state (incubation);
≫ `susceptibility`: Susceptibility to an infection by age group;

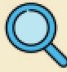

## Interventions Parameters

≫ `isol_pct_time_series`: A daily percentage of citzens that stayed at home;
≫ `quarantines`: Configurations relative to quarantines and hospitalizations of severe particles;
≫ `diagnostics`: Configurations relative to available diagnostics to the population;

**Fig 2. Most relevant parameters.** A non exhaustive classification of parameters used for a COMORBUSS simulation. Further description of parameters can be found on Section 5 in S1 File, while a complete list of parameters and their values can be found in the Git repository.

**Modeling disease.** Each agent is characterized by its age, which determines the agent's susceptibility, probability of developing symptoms and probability of dying from the disease. When a susceptible agent encounters an infectious one (pre-symptomatic, asymptomatic, mildly or severely symptomatic), it has a probability of becoming exposed. After an incubation period, this agent becomes pre-symptomatic, and after an activation period, its state is converted to either asymptomatic, mildly or severely symptomatic. The distribution of these states is estimated empirically from actual statistics [13, 14]. After a random period, agents are converted to recovered (or dead), see Section 5 in S1 File.

**Vaccine modeling and effect.** As our aim is to evaluate transmission patterns under different mitigation strategies, we are naturally interested in vaccines that can also affect disease dynamics either by blocking virus infection or transmission. Unfortunately, studies of vaccine efficacy so far did not address these outcomes directly and we lack data for modeling these mechanisms. We argue, based on the results shown in Fig 3, that vaccines are important mechanisms for individual protection, and, in turn, they might also have an impact at the population level by decreasing transmission of the virus to susceptible individuals. However, this is a secondary effect: what really defines the trend for relative reduction of cases is the chosen

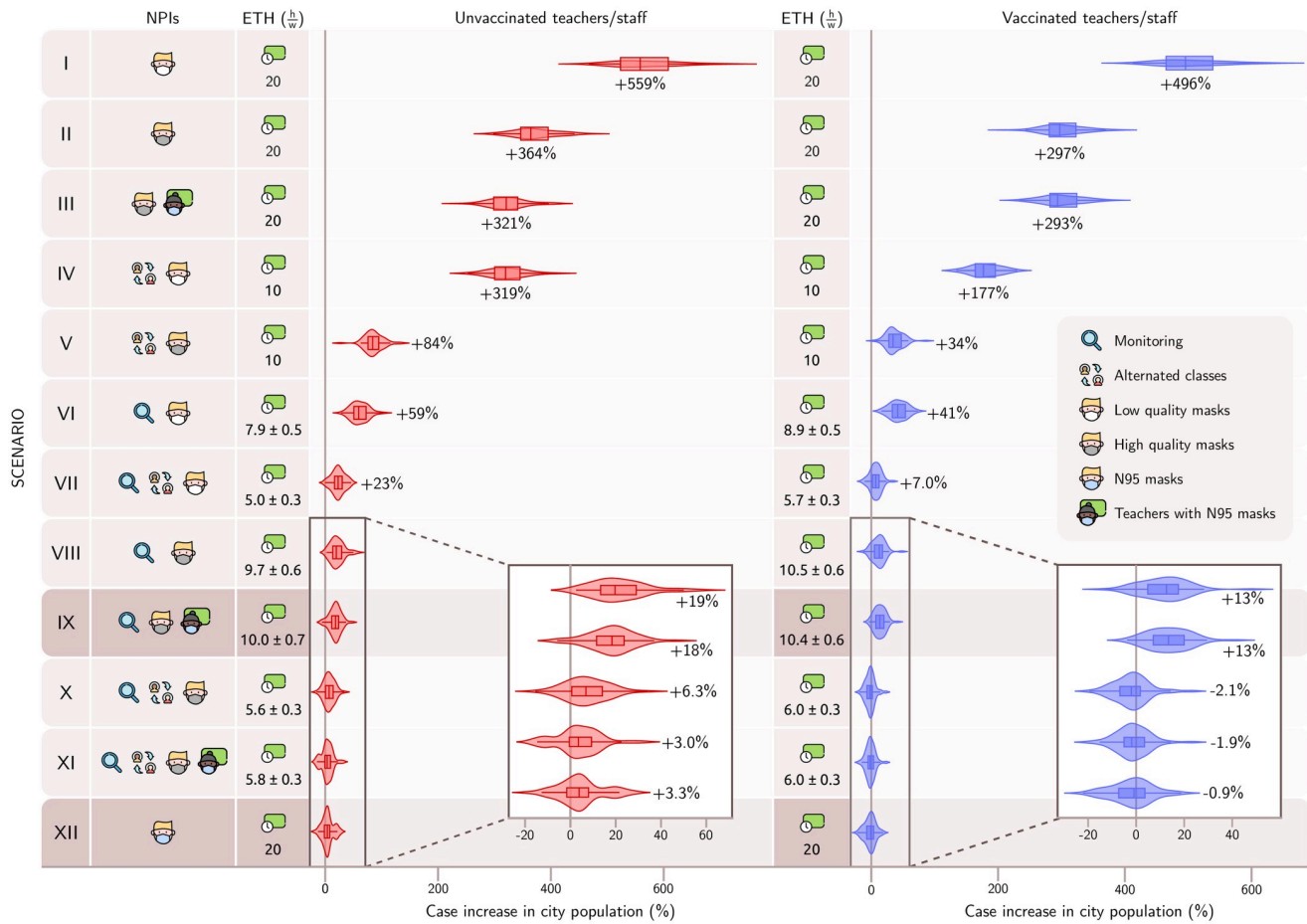

**Fig 3. Combination of NPIs measures in comparison to the baseline model settings.** Left panel: Case increase under different scenarios with unvaccinated teachers and staff. Right panel: Case increase under different scenarios with vaccinated teachers and staff. The effective teaching hours in hours/week $\frac{h}{w}$ and case increase in school population with respect to baseline are displayed for each NPIs combination. In case the active monitoring is also applied, the mean and standard deviation over 60 realizations for the effective teaching hours are shown. The proportional increase in the number of cases is displayed as violin plots (median, lower and upper quartiles), with kernel density estimates for distributions.

combination of NPIs. This is seen by comparing the worst and best-case scenarios regarding vaccination (left and right in Fig 3, respectively). We simulate a best-case scenario by initializing all school workers as perfectly vaccinated, such that their vaccination blocks all infection. We observe that the general picture remains unchanged within error bars, preserving almost the same ordering among scenarios. It is still unknown whether current vaccines can block infections. With all these and other unknowns in mind, we find it very reassuring that the two limiting cases we have studied are structurally similar and lead to sound conclusions regarding the effectiveness of school activity protocols.

Secondly, we investigate the effects of NPI adoption under different scenarios of partial vaccination for the general population (see Fig 4). Our main interest in this analysis is to evaluate the viability of the proposed measures for countries with different vaccination coverage, both in the well covered European continent and in the under-vaccinated African continent. We observe that the correct choice in NPIs can effectively protect the community even for low vaccination coverage, while poor adoption of NPIs can lead to high infection rates even for high vaccination coverage. Since we are dealing with larger segments of the population instead of just the school sub population, these simulations were performed with a more realistic vaccination model which only partially protects each agent with a biological efficacy of 98% for infection, resulting in an effective vaccine efficacy of 90% for the scenario where no NPIs are adopted. Although it tends to be more realistic, this model is highly complex to adjust and interpret because the measured vaccine efficacy is closely related to the running epidemiological scenario which responds to the adopted NPIs [15–17].

**Modeling services.** The city infrastructure is modeled by creating individual instances for each service (schools, hospitals, markets, restaurants, shops etc.) and by assigning agents to work/visit that location if they belong to an appropriate age group (a child may not work at a

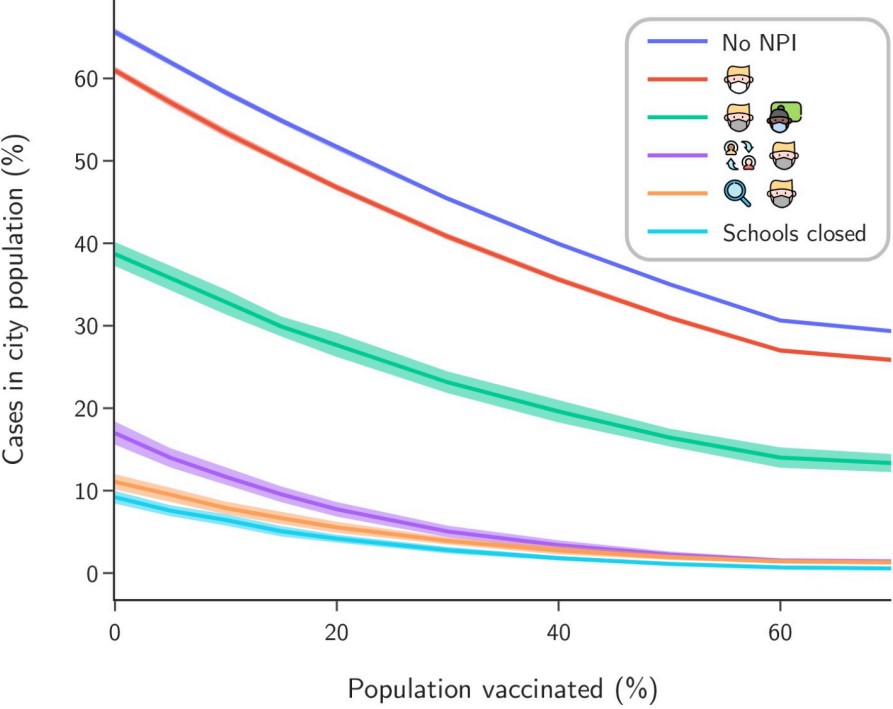

**Fig 4. Population fraction infected at the end of the simulation period (77 days) under varying vaccination coverage.**

shop, and an adult may not attend class). Worker agents are relocated to that service location during their shifts, while the visits of client agents are simulated stochastically. An hourly visitation rate of a service by an agent is empirically estimated, taking into account the service's opening hours and average visitation frequency of real clients; for details see Section 5 in S1 File. Additionally, agents may be allocated as guests to special services, which implies that their standard location is changed from their homes to that service instance. In this way we distinguish between hospitalizations, hotel quarantines and nursing home patients.

A novel point of our model is the creation of contact networks contextualized by social activity. The ratio of encounters between workers and clients as well as the clustering properties of a contact network is naturally dependent on the observed social context. For example, in restaurants there is a clustering of clients belonging to the same table, and contact between different tables is mediated by the contact of a shared waiter. Contact networks in schools, hospitals, stores etc., are all considerably different from each other. COMORBUSS updates random contact networks every hour for all the agents in the service instances, while respecting the characteristic architecture of the contact network of that type of service and distinguishing between the social roles of agents. Details and examples may be found in Section 5 in S1 File.

## Model calibration and closed schools as baseline

We aggregate socio-geographical data, as well as epidemiological data to COMORBUSS from May 9th 2020 to July 25th 2020, and leave the infection probability $p$ and the mean number of contacts $c$ in the City Hall to be calibrated using the empirical measure $\hat{v}$ obtained from the inference of states from data (see Section C). For a given $y = (q, d) \in [0, 1] \times \mathbb{R}_+$ we denote by $\hat{\mu}^y$ the empirical measure given by 400 independent realizations of COMORBUSS with $p$ and $c$ chosen as $(p, c) = y$. We construct an estimate $\hat{x}$ for $x = (p, c)$ by minimizing over all $y$ the $L_1$-Wasserstein distance between $\hat{v}$ and $\hat{\mu}^y$, see Section 6 in S1 File.

We initialize the community according to its demographics and household distribution, see Section 6 in S1 File. The disease state of agents is proportional to the average inferred epidemiological data for day May 9th 2020. The calibrated model is in excellent agreement with the estimated data and we use it as a baseline. This scenario resulted in an average of 3007±249 new infections in the population, in which 25% of those infections occurred in the school population, a measure that will serve as a baseline for keeping schools open in study cases.

## Poorly ventilated classrooms

In poorly ventilated classrooms, the main transmission mechanism is by aerosols emitted by an infected agent. The aerosols can remain suspended in the air, thereby reaching agents far from the original emitters [18, 19]. To model this exhaled air without reference to the microscopic pathogen concentration, we follow the exposition in [20, 21], describing the evolution of concentration of quanta in a closed space. *Quanta, introduced by Wells, measure the expected* rate of disease transmission, interpreted as infection quanta transference between pairs of infected and susceptible agent [22].

In our model, we denote by $C$ ($quanta/m^3$) the total concentration of quanta inside a classroom of volume $V$. Classrooms contain a total of $N$ agents, with $S$ susceptible individuals, $I_s$ infected students and $I_t$ infected teachers. All breathe uniformly at a rate $B = 0.5$ $m^3/h$. Since mask wearing can decrease the amount of aerosols emitted to the air, we denote for each agent the penetration mask factor $p_m^i \in (0, 1)$, with $i = s, t$: see Fig 5.

Each person exchanges quanta with the air depending on breathing activity. We introduce the concentration of quanta expelled by students $C_s = 40$ ($quanta/m^3$) and teachers $C_t = 72$ ($quanta/m^3$) [21] (corresponding to voiced counting [23]). Under a well-mixed room

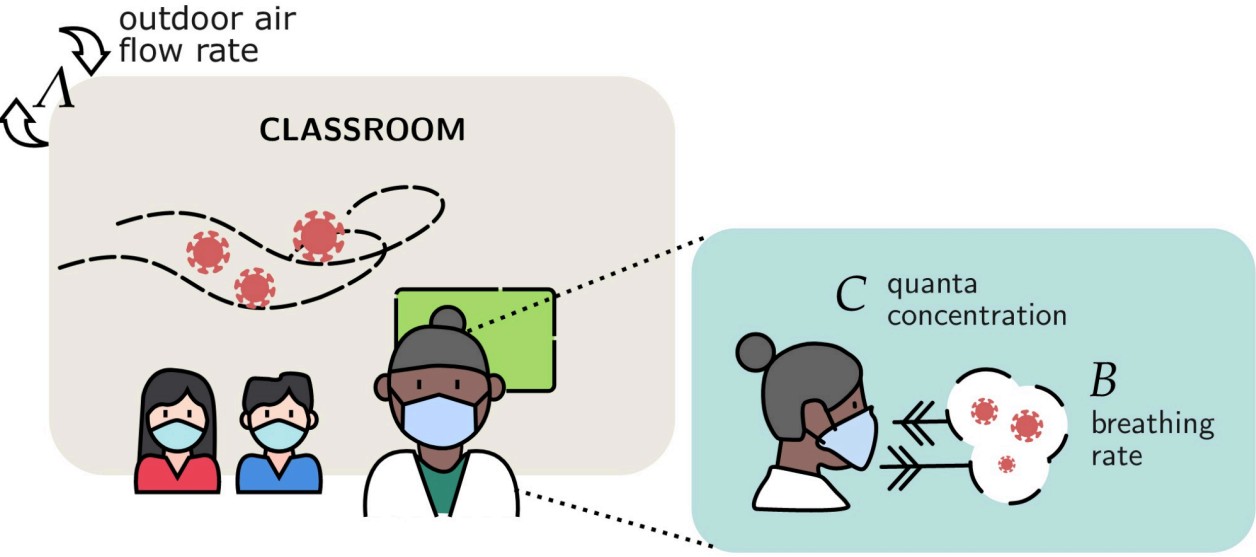

**Fig 5. Airborne transmission model inside school environment.** The classroom is an enclosed space in which airborne transmission has a high chance of occurrence. Contaminated particles are spread over the classroom, allowing long range infections. The fresh air rate flow $\Lambda$ quantifies the classroom ventilation. The quanta concentration $C$ varies in the environment depending on the breathing activity.

assumption, the total concentration of quanta $C$ ($quanta/m^3$) inside the classroom satisfies the mass equation:

$$V\frac{dC}{dt} = -(\Lambda V + NB)C + B(C_s I_s p_m^s + C_t I_t p_m^t). \tag{1}$$

Note that our setting relies on the fact that the airborne particles remain airborne before being extracted by the outdoor air flow $\Lambda$ (typically reported as air changes per hour or ACH) or inhaled by an agent. We investigate the poor ventilation limit $\Lambda = 0$ and fresh air flow ($\Lambda > 0$), see Section 7 in S1 File.

The amount of quanta inhaled by the $i$th agent inside the class over a time $t$ is the inhaled dose $D_i(t) = B p_m^i \int_0^t C(t)dt$. We evaluate this integral over the solution to Eq (1). Using the inhaled dose of each agent, we plug it into the Wells-Riley model to calculate the probability of a susceptible individual being infected [19, 20], which consists in estimating the risk of infection in indoor environments via

$$p_{\text{indoor}}^i(t) = 1 - e^{-r_i D_i(t)},$$

where $r_i$ is the relative susceptibility (an age-based measure [24]) for the agent $i$. We set the relative susceptibility of children (aged 0 years to 14 yr), adults (aged 15 yr to 64 yr) and the elderly (over 65 yr) to $r_i = 0.23, 0.68, 1$, respectively. To determine the source of infection of a particular exposed individual, we pick a random individual uniformly from all of the infectious individuals in the enclosed space, see Section 7 in S1 File.

## Results and discussion

We present three classes of results, each with their own implications to health protocol design: i) effectiveness analysis of a large set of protocols; ii) analysis of how the most relevant protocols depend on good mask practices and ventilation; iii) predictions on protocol effectiveness when challenged by more infectious viral strains.

It must be noted that, while our model can be easily applied to other communities via our systematic data integration procedure, acquiring good quality datasets and ensuring their compatibility is the most limiting challenge in our methodology. For example, we have found that in many cities the census data and the database describing the available services are offset by a few years. We had the experience of modeling cities which had explosive growth during those years and these two datasets became so incompatible that there were not enough agents from the demographic data to work on the most recent infrastructure. We naturally need to rely on interpolation and extrapolation of historic datasets in such cases. Regardless, we find that a close collaboration with city managers, as we had in Maragogi, is ideal for ensuring the quality of the data and as well as in identifying trends and supporting modeling choices. This is critical in order to evolve the model as we learn more about the disease and the social behavior also changes in response to it. We document our experiences on this process in the first four sections of the S1 File.

## NPIs and vaccination

Across 27 schools, the total school population is 8,528, with 7,557 students. We quantified the effects of five NPIs on the school population, which consists of teachers, school staff and students. Each NPI is described in Fig 6. Although there is still controversy in the literature about the efficiency of surgical masks for filtering particles [25] and side-effects [26], we assign mask quality via their permeability factors $p_m$, as indicated in Fig 6.

We simulate school activities with different NPI and compute the percentage increase of cases with respect to the baseline. The results are presented in Fig 3 along with the effective teaching hours. Conducting classes in full shift and wearing only poor quality masks leads to a 559% increase in infections. We note that the wearing of N95 masks by teachers and staff is particularly effective at reducing the number of cases compared to other scenarios, and we highlight this NPI in Fig 3 (darker color). Active monitoring curbs spreading, at the expense of the effective number of teaching hours.

We assume in the simulation that vaccinated teachers and staff are initialized with protective neutralizing antibodies against COVID-19. This blocks any possible infection chain starting from these individuals. The right panel in Fig 3 displays the effectiveness of NPI combinations with vaccinated employees. If employees are not vaccinated, case rates increase in all scenarios. The case increases in the highlighted (darker color) scenarios are reduced for both unvaccinated and vaccinated employees, indicating that they are a potential source of infection for the school population.

We also analyze the robustness of our results when considering a larger city, using as example the regional capital of Curitiba with almost 2 million inhabitants. We observe how bad protocols lead to sharp increase in infections while good ones successfully avoid this phenomena. Most remarkably, the relative effectiveness rank between intervention is preserved, even if the case increase relative to the baseline is less pronounced, see further details in Section 8 in S1 File. This not only shows the stability of the protocols but also indicates that smaller cities are more vulnerable and in need of appropriate protocols.

We also consider the effectiveness of NPI scenarios under different levels of vaccination coverage, see Fig 4. Our motivation is to asses the viability and safety of public health decisions even in countries with low coverage, such as African countries. In fact, even with low vaccination coverage, we find that a good choice of NPIs in schools also protects the wider community better. At the same time, poorly chosen or non-existent NPIs may leave the communities highly exposed, regardless of vaccination coverage. We therefore stress the importance of appropriate NPIs and protocols, whether or not the underlying country enjoys good vaccine

| NPI | Description |
|---|---|
| Reduced Workload | Schools function with shifts of two hours instead of four hours. |
| Alternating Groups | Schools function with reduced class sizes, and in particular classes are separated into 2 groups having in-person activities on alternate days. |
| Use of Mask | Low quality: $p_m = 0.5$ — Good quality: $p_m = 0.3$ — N95 or PFF2: $p_m = 0.05$ — Teachers and staff with N95. |
| Active Monitoring | Schools function under the following measures:<br><br>• Symptomatic people are tested;<br><br>• If a case is found in a classroom, their activies are suspended for 14 days;<br><br>• Students are tested and isolated (14 days) when they are symptomatic or a family member is confirmed positive;<br><br>• Teachers which had contact with a classroom in which there were confirmed cases are tested and suspended for 14 days in the case of positive result;<br><br>• School is closed for one week if there are two cases in distinct classes within a week. |

**Fig 6. NPIs description.** The icons distinguish the non-pharmaceutical interventions evaluated in this study. In scenarios involving masks, the mask penetration factor $p_m$ is uniform for all individuals, except for teachers wearing PFF2 masks.

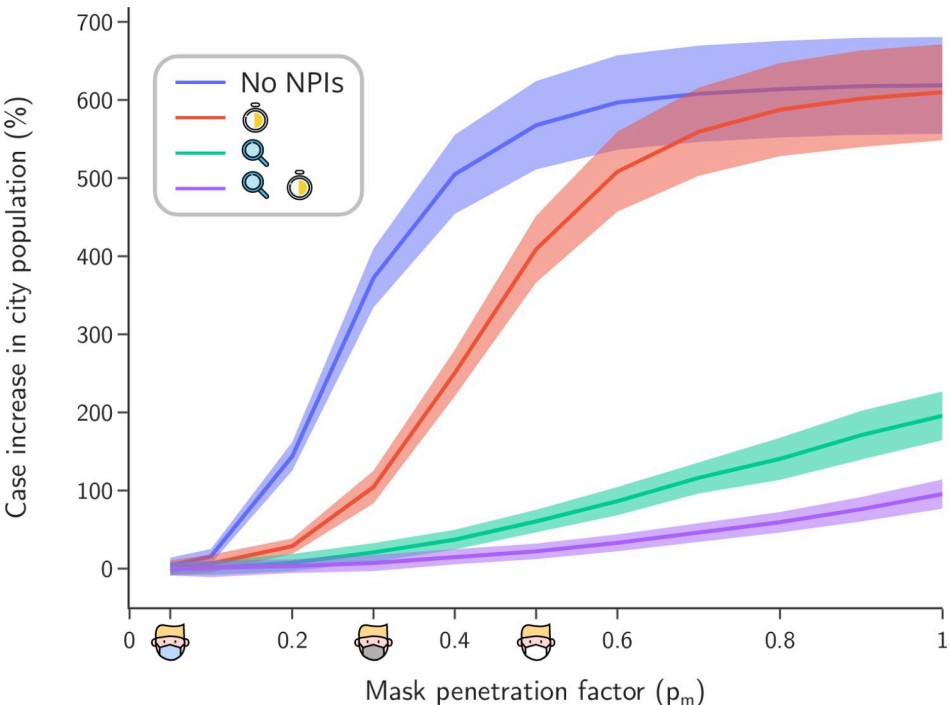

**Fig 7. Sensitivity analysis across mask penetration factor $p_m$.** Cases increase in school population (solid lines) versus the mask penetration (mean values over 60 realizations for each $p_m$ value).

coverage. We recall that the cities are modelled with only essential services operating, including schools. Lessons drawn here extend to other services and social contexts to avoid the worsening of outbreaks.

## Sensitivity analysis: Mask penetration and ventilation

We quantify the relevance of the mask penetration factor $p_m$ and ventilation air flow rate $\Lambda$ for the increase of COVID-19 cases in the cities. Assuming that all pupils wear masks with the same $p_m$, Fig 7 shows the impact of the penetration factor on the number of cases if schools are kept open. Results are sensitive to the penetration factor of the masks, as seen by comparing the first (poor quality or practices, $p_m = 0.5$) and second (high quality masks, $p_m = 0.3$) simulation scenarios, showing a decrease of almost 200% in cases regardless of the vaccination status of employees. We also observe that the use of N95 masks by employees increases the effective teaching hours in the scenarios with active monitoring.

Fig 8 shows the sensitivity analysis when the ventilation rate is varied inside classrooms. Based on recomendations by the American Society of Heating, Refrigerating and Air Conditioning Engineers (ASHRAE) [27], we calculated the minimal ventilation rate of $\Lambda_1 = 0.8 \; h^{-1}$ for unoccupied classrooms using their average dimensions in Maragogi. Ventilation rates for half full and full classrooms are $\Lambda_2 = 3.8 \; h^{-1}$ and $\Lambda_3 = 6.6 \; h^{-1}$, respectively; for further details see Section 7 in S1 File.

## Scenarios with more infectious variants

In investigating the effectiveness of school safety protocols during infection waves caused by new, more infectious variants, we are drawn to the limiting worst case scenarios. As such, we

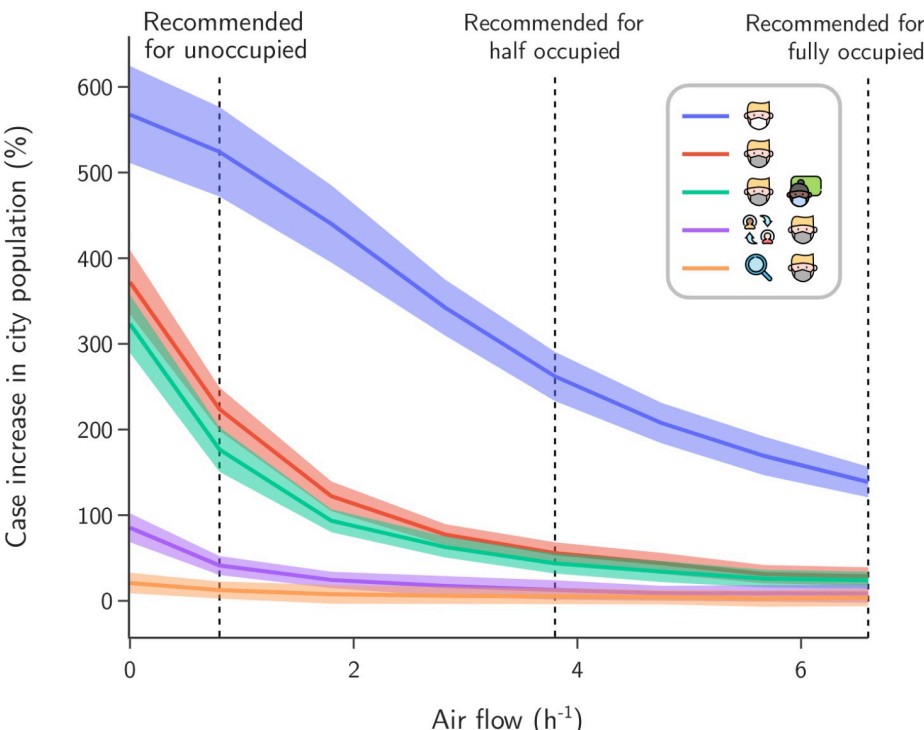

**Fig 8. Sensitivity analysis across ventilation Λ.** Cases increase in school population (mean and standard deviation) as a function of classroom ventilation rate. Dashed lines indicate the recommended ventilation rates: $\Lambda_1 = 0.8\ h^{-1}$ (unoccupied room), $\Lambda_2 = 3.8\ h^{-1}$ (half occupied room) and $\Lambda_3 = 6.6\ h^{-1}$ (fully occupied room), following ASHRAE standard for an average classroom in Maragogi.

assume that the new variant completely avoids the acquired immunity from vaccination or previous infections. New variants are modeled by an increase in the population susceptibility, therefore encompassing both our contact and aerosol transmission models. Susceptibility is increased by the multiplying factor over all age groups as a limiting case.

The results are depicted in Fig 9. As expected, the total population infected increases monotonously with the increase in susceptibility, with poor protocols for school activities leading to extreme infection rates across the community. Most importantly, not only good protocols still lead to remarkable decrease in infection rates but the relative rank of effectiveness between protocols is preserved regardless of how much susceptibility is increased. This shows the stability of good protocols and makes the point that their adoption should always be a top priority even when facing new potentially variants.

## Conclusion

The airborne transmission mechanism of COVID-19 is the main cause of infections in school environments in classrooms with poor air circulation. Since many classrooms are equipped with air conditioning or heating, most have poor air circulation. Therefore, reducing the class size does not necessarily curb spreading because an infected person can emit aerosols that stay in the air and infect students far away in the same classroom.

Vaccination of employees is an essential measure. Still, in the absence of other measures such as monitoring and quarantines, the number of cases in the cities is likely to increase by 177% if only the use of low quality masks and alternated classes are implemented.

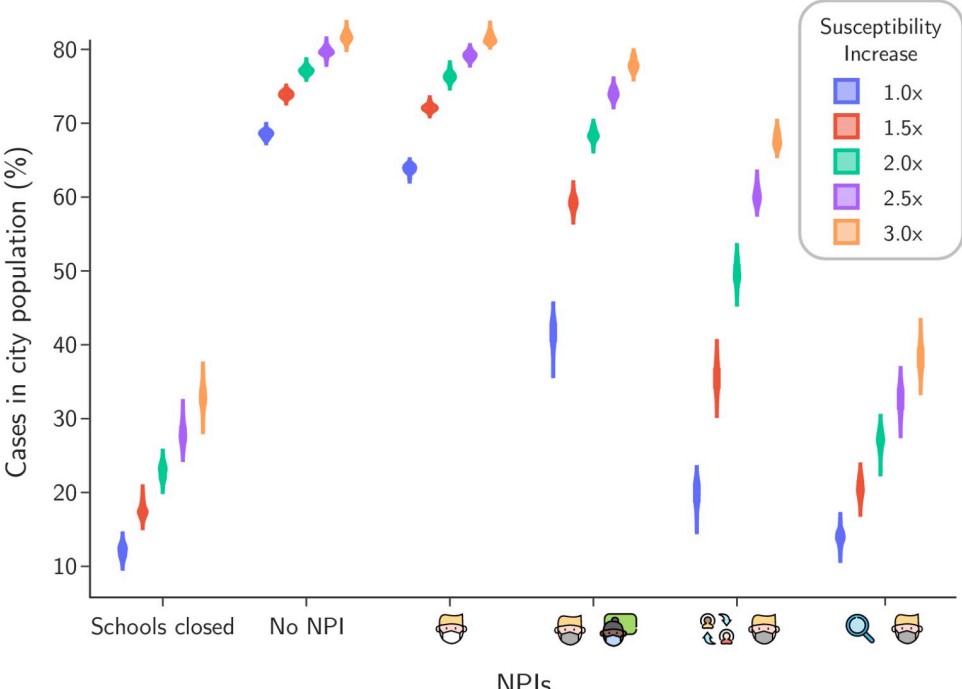

**Fig 9. Population infected in case of increase in susceptibility.** For each intervention scenario, we show the distribution in the percentile of the population infected provided the susceptibility of the population is increased uniformly by a multiplying factor.

The penetration factors provided by manufactures and used in our simulations are idealized. In practice, the fit of a mask and the practices of users result in lower filtration efficacy. Indeed, after testing a model of contagion based on a study of Canadian classrooms [28], we compared the ensuing results with our own aerosol model under the same class conditions but varying penetration factors, in order to estimate its value in these classrooms. We were alarmed to find that the effective penetration factor for the Canadian classrooms in that study was only 0.5, despite the assumption of high quality masks. It would therefore be of great benefit to instruct the general population in proper mask use. Otherwise, the potential effectiveness of sanitary protocols will be compromised as the achieve penetration factor increases (Fig 7).

All these findings can be explained by three facts: teachers are more susceptible than children, they expel more virulent particles since they are constantly speaking loudly and they are the most effective bridges of transmission between isolated classes. Therefore, high quality masks not only protect the individual teacher, but also suppress community infection.

Our most striking result is that one must adopt the appropriate NPIs and behavioral protocols in order to safely continue school activities during a pandemic, regardless of vaccination coverage. Good protocols can protect countries even with poor vaccine coverage. Conversely, bad protocols may seriously aggravate the underlying public health crises even in countries with very high vaccination coverage. This is in great part due to the long duration of social contacts in schools, easily leading to breakthrough infections without proper protocols. This is particularly relevant given that in many countries children are not routinely vaccinated for COVID-19, or when preparing for the emergence of new variants with potentially low cross immunity.

There is no single solution to a pandemic, but we draw hope in showing that the proper combination of NPIs, vaccination and behaviors permit the safe continuation of activities as fundamental and important as teaching.

## Supporting information

**S1 File. Detailed description of data collection, data analysis, COMORBUSS software, calibration, and sensibility analysis.**
(PDF)

## Acknowledgments

We acknowledge the City Hall of Maragogi (Prefeitura Municipal de Maragogi) and its Mayor Fernando Sérgio Lira Neto for establishing a technical cooperation with our team and providing the requested data. The graphics used to prepare the Figures were extracted from FreePick.

## Author Contributions

**Conceptualization:** Juliano Genari, Guilherme Tegoni Goedert, Dan Marchesin, Claudio J. Struchiner, Tiago Pereira.

**Data curation:** Sérgio H. A. Lira, Krerley Oliveira, Allysson Lima, José Augusto Silva, Hugo Oliveira, Maurício Maciel, Claudio J. Struchiner.

**Formal analysis:** Guilherme Tegoni Goedert, Claudio J. Struchiner, Tiago Pereira.

**Funding acquisition:** Claudio J. Struchiner, Tiago Pereira.

**Investigation:** Juliano Genari, Guilherme Tegoni Goedert, Sérgio H. A. Lira, Lucas Resende, Edmilson Roque dos Santos, Claudio J. Struchiner, Tiago Pereira.

**Methodology:** Juliano Genari, Guilherme Tegoni Goedert, Sérgio H. A. Lira, Krerley Oliveira, Lucas Resende, Edmilson Roque dos Santos, Claudio J. Struchiner, Tiago Pereira.

**Project administration:** Claudio J. Struchiner, Tiago Pereira.

**Resources:** Claudio J. Struchiner, Tiago Pereira.

**Software:** Juliano Genari, Guilherme Tegoni Goedert, Sérgio H. A. Lira, Krerley Oliveira, Adriano Barbosa, Ismael Ledoino, Lucas Resende.

**Supervision:** Guilherme Tegoni Goedert, Dan Marchesin, Claudio J. Struchiner, Tiago Pereira.

**Validation:** Sérgio H. A. Lira, Ismael Ledoino, Lucas Resende, Edmilson Roque dos Santos.

**Visualization:** Krerley Oliveira, Adriano Barbosa, Lucas Resende, Edmilson Roque dos Santos.

**Writing – original draft:** Guilherme Tegoni Goedert, Sérgio H. A. Lira, Krerley Oliveira, Edmilson Roque dos Santos, Claudio J. Struchiner, Tiago Pereira.

**Writing – review & editing:** Juliano Genari, Guilherme Tegoni Goedert, Sérgio H. A. Lira, Krerley Oliveira, Edmilson Roque dos Santos, Dan Marchesin, Tiago Pereira.

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
