## [Decision Letter · Decision Letter 0]

29 Jun 2022

PONE-D-22-13978Quantifying protocols for safe school activitiesPLOS ONE

Dear Dr. Pereira,

Thank you for submitting your manuscript to PLOS ONE. After careful consideration, we feel that it has merit but does not fully meet PLOS ONE’s publication criteria as it currently stands. Therefore, we invite you to submit a revised version of the manuscript that addresses the points raised during the review process.

We look forward to receiving your revised manuscript.

Kind regards,

Shinya Tsuzuki, MD, MSc

Academic Editor

PLOS ONE

Journal Requirements:

Additional Editor Comments:

Please answer each comment raised by the reviewers.

Reviewers' comments:

Reviewer's Responses to Questions

**Comments to the Author**

1. Is the manuscript technically sound, and do the data support the conclusions?

Reviewer #1: Yes

Reviewer #2: Yes

2. Has the statistical analysis been performed appropriately and rigorously? 

Reviewer #1: Yes

Reviewer #2: Yes

3. Have the authors made all data underlying the findings in their manuscript fully available?

Reviewer #1: Yes

Reviewer #2: Yes

4. Is the manuscript presented in an intelligible fashion and written in standard English?

Reviewer #1: Yes

Reviewer #2: Yes

5. Review Comments to the Author

Reviewer #1: This is an excellent research topic that investigates factors of COVID-19 virus transmission in schools using a variety of data sources. Findings from this study would be used as evidence-based to prevent COVID-19 in schools. A few comments to improve this article:   

1. Paragraph 32-54 is not suitable/ appropriate in the introduction. The introduction should focus on study background rather than explaining methodology and summarising study findings.

2. Results and discussion were combined in this article. suggested separating according to IMRAD. I find it a bit mixed up here, and the authors need to discuss or elaborate more based on study findings. 

3. Multiple methods and analysis were used to yield study findings from this study. I would suggest there is a paragraph that lists out all the variables and explicitly explains the variables used before the methods in the analysis.

Reviewer #2: I appreciate the importance of the results. I consider that the article has a high potential for impact.

My main concern is in such analyses, it typically is possible to express the optimization problem in multiple ways with similar outcome. I think is very important to consider the deviation between the current practice and the solutions provided.

Introduction

- The introduction should avoid nominalizations.

- The introduction presents a summary of the project results, this can be deleted.

Methods

- the methods have gaps in Agent based modeling

- The authors should be clearer about COMORBUSS.

Discussion

- The authors should include the limitation of study.

6. PLOS authors have the option to publish the peer review history of their article (what does this mean?). If published, this will include your full peer review and any attached files.

Reviewer #1: No

Reviewer #2: **Yes: **Daniel Henrique Bandoni

---

## [Author Response · Author response to Decision Letter 0]

22 Jul 2022

Dear Editor, 

We thank you and the referees for their constructive advice and the time dedicated to helping improve our manuscript. We addressed the concerns raised and improved the manuscript acoordingly. 

Please find the reply point-by-point raised comments below.

Sincerely Yours 

The Authors

Reviewer #1

> This is an excellent research topic that investigates factors of COVID-19 virus transmission in schools using a variety of data sources. Findings from this study would be used as evidence-based to prevent COVID-19 in schools.

Thank you for your kind words and for recognizing the value of our work.

> 1. Paragraph 32-54 is not suitable/ appropriate in the introduction. The introduction should focus on study background rather than explaining methodology and summarising study findings.

We follow the referee suggestion while keeping lines 51-54 at the end of the introduction. As it serves as a hook for readers.

> 2. Results and discussion were combined in this article. Suggested separating according to IMRAD. I find it a bit mixed up here, and the authors need to discuss or elaborate more based on study findings.

We present classes of results, each based on different experiments and their peculiarities

I) Exploration of the efficacy of different protocols;

II) stability analysis of the protocols varying conditions for airflow and mask quality;

III) stability analysis of the protocols when faced with more infectious viral strains. 

By combining the Results and the Discussion sections, we are able to present each class of results and immediately explores their consequences. This choice keep the focus of each class of results and highlight their individual consequences to the design of public health policies. 

We hope to have clarified our reasons and that you can support this choice. 

> 3. Multiple methods and analysis were used to yield study findings from this study. I would suggest there is a paragraph that lists out all the variables and explicitly explains the variables used before the methods in the analysis.

In Figure 2 we show the 4 types of parameters present in our model. Larger lists are still found throughout SI section 5 and the complete list is available in the repository.

Reviewer #2

> I appreciate the importance of the results. I consider that the article has a high potential for impact.

We appreciate your evaluation of the importance of our work.

> My main concern is in such analyses, it typically is possible to express the optimization problem in multiple ways with similar outcome. I think is very important to consider the deviation between the current practice and the solutions provided.

We agree that optimization depends sensitively on the constrains. In our study, optimization plays a role in the calibration of the epidemiological model from real world data. Then, our results do not optimize on the protocols to obtain a low number of infections. Rather, we analyse the outcome of multiple protocols that could be applied in schools. 

> Introduction

- The introduction should avoid nominalizations.

- The introduction presents a summary of the project results, this can be deleted.

This has been corrected by removing lines 32-50. We also believe that striking out this denser part of the text reduces the nomizalization and improves the overall readability as you intended.

> Methods

- the methods have gaps in Agent based modeling

- The authors should be clearer about COMORBUSS.

We have improved on that in lines 112-135 and strengthened the link between the main text and the Supplementary Information. 

> Discussion

- The authors should include the limitation of study.

Our largest limitation are the challenges involved in acquiring datasets of sufficient quality and their integration to COMORBUSS to produce good and reliable models. Following your suggestion, we present the most important problems in the main text (lines 243-354) and reinforced its integration to the SI.

---

## [Decision Letter · Decision Letter 1]

9 Aug 2022

Quantifying protocols for safe school activities

PONE-D-22-13978R1

Dear Dr. Pereira,

We’re pleased to inform you that your manuscript has been judged scientifically suitable for publication and will be formally accepted for publication once it meets all outstanding technical requirements.

Kind regards,

Shinya Tsuzuki, MD, MSc

Academic Editor

PLOS ONE

Additional Editor Comments (optional):

Please check your proof carefully according to the comments by reviewer 1.

"As for the language in this paper, I found some flaws that are more likely to be due to carelessness than actual grammatic errors, but I want to state that as a non-native English speaker I don't feel qualified to judge the language."

Reviewers' comments:

Reviewer's Responses to Questions

**Comments to the Author**

1. If the authors have adequately addressed your comments raised in a previous round of review and you feel that this manuscript is now acceptable for publication, you may indicate that here to bypass the “Comments to the Author” section, enter your conflict of interest statement in the “Confidential to Editor” section, and submit your "Accept" recommendation.

Reviewer #2: All comments have been addressed

2. Is the manuscript technically sound, and do the data support the conclusions?

Reviewer #2: (No Response)

3. Has the statistical analysis been performed appropriately and rigorously? 

Reviewer #2: Yes

4. Have the authors made all data underlying the findings in their manuscript fully available?

Reviewer #2: Yes

5. Is the manuscript presented in an intelligible fashion and written in standard English?

Reviewer #2: Yes

6. Review Comments to the Author

Reviewer #2: The authors have taken into account the comments and suggestions made by the reviewers. The end article is a good work on an interesting topic. I have no further comments.

7. PLOS authors have the option to publish the peer review history of their article (what does this mean?). If published, this will include your full peer review and any attached files.

Reviewer #2: No

---

## [Editor Report · Acceptance letter]

2 Sep 2022

PONE-D-22-13978R1 

Quantifying protocols for safe school activities 

Dear Dr. Pereira:

I'm pleased to inform you that your manuscript has been deemed suitable for publication in PLOS ONE. Congratulations! Your manuscript is now with our production department. 

Kind regards, 

on behalf of

Dr. Shinya Tsuzuki 

Academic Editor

PLOS ONE